# A Study on the Relationship between Internet Overdependence and Anger Response among Young Adults during COVID-19 Pandemic: Moderating Effect on Negative Emotions

**DOI:** 10.3390/ijerph20032435

**Published:** 2023-01-30

**Authors:** Sun Kyung Kang, Jin Kwon, Kwanghyun Kim

**Affiliations:** 1Department of Social Welfare, Sogang University, Seoul 04107, Republic of Korea; 2Department of Social Welfare, Yemyung Graduate University, Seoul 06723, Republic of Korea; 3Department of Social Welfare, Seoul National University, Seoul 08826, Republic of Korea

**Keywords:** COVID-19, young adults, negative emotion, internet dependence, anger responses

## Abstract

The aim of this study is to examine how Internet dependence affects anger responses during the COVID-19 pandemic. Owing to social distancing policies, Internet dependence has intensified, and the prevalence of anger has significantly increased. To understand this phenomenon and draw some implications, the “frustration–aggression hypothesis” was utilized for the theoretical framework and anger response was categorized into functional and dysfunctional anger responses. An analysis shows that overdependence on the Internet has a positive effect on the dysfunctional anger response. At the same time, other negative emotions replace anger, reducing the possibility of a dysfunctional anger response. Accordingly, a need for a constant effort to understand the circumstances of the young generation living in the “new normal” is emphasized; moreover, this paper suggests some theoretical and practical implications.

## 1. Introduction

The COVID-19 pandemic had an emotional, behavioral, and psychological impact on various age groups all around the world [1]. People experienced the fear of social isolation and economic uncertainty as well as the fear of infection and death [2]. Addictive behaviors and violence increased during the pandemic [3,4]. The risks of psychological disorders (including anxiety disorders, depression, post-traumatic stress disorder (PTSD), obsessive-compulsive behaviors, panic, and paranoia) were intensified due to the pandemic [5]. However, policies that focused on the infectious aspect of the pandemic overlooked emotional difficulties such as fear, anger, and depression among individuals [6].

In particular, previous research [7] conducted during the COVID-19 pandemic with 2237 participants aged 16–75 years found that 56% of participants (n = 1255) experienced feelings of anger or broken relationships with others because of COVID-19. In a longitudinal survey in South Korea concerning people’s emotional status during the pandemic, anxiety (60.2%) was the most prevalent emotion experienced by participants in the first survey conducted in February 2020, followed by fear (16.7%), shock (10.9%), and anger (6.7%) in order. However, in a second survey conducted in March 2020, the prevalence of anxiety had decreased to 48.8% while the prevalence of anger (21.6%) had increased 2.2-fold [8]. These findings demonstrate that fear of the pandemic changed into anger.

Anger, a natural human emotion, can have positive effects such as mobilizing psychological resources, promoting patience, and protecting self-esteem [9]. However, it can sometimes also result in negative outcomes such as physical and mental health problems [10], social relationship problems [11], and self-destructive behaviors [12]. Prior studies suggest that these differences in outcomes are determined by the types of anger responses. In particular, some categorize anger responses into two types: functional and dysfunctional anger responses [13,14,15,16]. Unfortunately, there is no consensus on the definitions of functional and dysfunctional anger responses. Nevertheless, after considering previous definitions, in this study, a dysfunctional anger response is defined as expressing anger violently toward others or avoiding emotion, while a functional anger response is defined as deliberating and expressing emotion nonviolently.

There have been various studies on increased anger during the COVID-19 pandemic. In one study [7], younger age, greater likelihood of experiencing financial difficulties, greater perceived risk of COVID-19, and acquiring information about COVID-19 from social media were strongly related to experiencing anger. In another study [17], Twitter users who posted frequently and vigorously were found to be more likely to demonstrate anger online during the pandemic. Despite other explanations for this increase in anger during the pandemic, it is reasonable to assume (based on prior studies) that the phenomenon is somehow related to online activities.

Under social distancing policies, the prevalence of Internet use and Internet-based addictive behaviors, such as Internet addiction, online gaming disorder, online gambling disorder, pornography addiction, and smartphone-use disorder, has soared in several countries [18,19]. Even though these changes are not limited to certain groups, young adults who were already familiar with the Internet and vulnerable to Internet dependence since before the pandemic also suffered from increased dependence on the Internet during the pandemic [20].

The positive functions of the Internet enhance convenience in people’s lives, greatly enhancing the development of smooth online networks. By contrast, considerable concerns about the dysfunctional aspects of the Internet have also been suggested. Previous research [21,22,23] has demonstrated that excessive use of the Internet heightens the dysfunctional expression of anger. While a number of terms such as Internet addiction, Internet addiction disorder, pathological Internet use, problematic Internet use, and compulsive Internet use have been used to describe excessive Internet use in prior studies, no consensus on such terms has emerged [24]. In this study, the term “Internet overdependence” is utilized because this term is more neutral than diagnostic terms containing words such as disorder or addiction, which more intuitively demonstrate the phenomenon.

According to the “frustration–aggression hypothesis,” aggression is a consequence of frustration [25]. Moreover, frustration is defined as “an interference with the occurrence of an instigated goal-response at its proper time in the behavior sequence (p. 7)” [26]. In the original model, two assumptions were proposed: (1) all frustration triggers an impulse to aggression that tries to harm others, and (2) all aggressive behavior stems from past setbacks. However, these assumptions were criticized for being based on an excessively mechanistic and simplified perception [25,27]. In this context, Berkowitz [27] reformulated the model of frustration and aggressive behaviors. With empirical evidence, he assumed that “the aggressive consequences of frustration might not be apparent unless aggression-facilitating cues were also present in the immediate situation (p. 65).” In other word, a frustrating event may only result in aggressive inclination, but other internal or external cues can trigger aggressive behavior [28]. Berkowitz also suggested that a negative affect (e.g., anger, sadness, anxiety, and fear) may appear after frustration and before aggressive behavior, depending on personal attributions and environmental conditions [29]. The type of negative affect appearing based on various internal and external factors will determine a “fight-or-flight response” to frustration [28]. This more elaborated idea of frustration has been empirically tested in other studies [30,31].

Internet overdependence can be a signal that captures the frustration cue that triggers negative emotions resulting in aggressive anger responses. First, those who are Internet-dependent are likely to experience self-frustration. By excessively depending on Internet activities, real-life personal or social goals are often frustrated [32]. After experiencing frustration, Internet-dependents often blame themselves for their failure and become increasingly obsessed with the online world [32,33,34]. Moreover, perceived enjoyment, envy, information overload, and social overload provoked by using online social network services can be further sources of frustration [35].

In this context, a hypothesis was established that young adults who suffer from Internet overdependence may respond more dysfunctionally and aggressively toward anger. Furthermore, according to Berkowitz’s reformulated “frustration and aggression hypothesis,” anger can be combined with or replaced by other negative emotions. In this study, therefore, negative emotions excluding anger were applied as moderating variables.

## 2. Materials and Methods

### 2.1. Study Model and Hypothesis

The model for this study is presented in Figure 1. First, the independent variable is overdependence on the Internet, and the dependent variable is anger response. Anger response was analyzed as two sub-factors: dysfunctional anger response and functional anger response. The other negative emotions—sadness, anxiety, feeling abandoned, and feeling isolated—are moderating variables. To test the moderating effect, gender, age group, educational background, monthly income, and marital status were used as control variables.

### 2.2. Measures

The independent variable of this study—”overdependence on the Internet”—was measured using 15 questions from the Adult Internet Addiction Self-Diagnosis Simple Scale (KS-A) developed by the National Information Society Agency. Overdependence on the Internet (measured using a 4-point scale) consisted of four sub-variables: daily life disorder, virtual world orientation, withdrawal, and resistance. A higher score denoted greater Internet independence. The reliability coefficient value of this scale was shown to be 0.747. The dependent variable “anger responses” was measured using the Korean version of the Behavioral Anger Response Questionnaire (K-BARQ), which consists of six sub-factors: direct anger out, assertion, support-seeking, diffusion, avoidance, and rumination. The overall internal consistency reliability (Cronbach’s Alpha) of BARQ was 0.75 [36], and that of K-BARQ was 0.80 [37]. The anger response variable was measured using a 5-point scale: the higher the score, the higher the level of either dysfunction or function. The dysfunctional anger response (conceptually discussed and constructed in this study) consists of three sub-factors: direct anger out, avoidance, and rumination; the functional anger response consists of three sub-factors: assertion, support-seeking, and diffusion. The reliability coefficient value of the dysfunctional anger was 0.754, while that of the functional anger response was 0.810. Finally, “negative emotion” was measured using five questions, with the response to each question being either yes (1) or no (0). The sum of all questions was calculated.

### 2.3. Study Procedure and Data Analysis

The data used in this study were derived from the Addictive Behavior Survey of the Institute for Life and Culture of Sogang University: the dataset for an online survey targeting adult men and women in their 20s and 30s who reside throughout South Korea. The survey was conducted from 28 August 2021 to 18 September 2021, using the proportional allocation and convenience sampling method through an online professional survey company. Responses were excluded that were untrustworthy, insincere, excessively quick, and containing logical error, resulting in a final sample of 762 cases for analysis.

The statistical analysis used in this study was conducted as follows. First, to examine the characteristics of the sample, demographic and sociological characteristics were analyzed using a frequency analysis. Next, a descriptive analysis was conducted to examine the mean, standard deviation, kurtosis, and skewness of major variables. In sequence, after analyzing the correlation between variables, an analysis of the moderating effect was finally conducted. The SPSS version 23.0 statistical software package was used for the analysis, and the moderating effect was determined using PROCESS MACRO.

## 3. Results

### 3.1. Demographic and Sociological Characteristics of Subjects

After analyzing the general characteristics of the participants, it was found that 49.7% (n = 379) were males and 50.3% (n = 383) were females, which means that there was no significant difference according to gender. By age, 47.9% (n = 365) were in their 20s and 52.1% (n = 397) were in their 30s, which means that there were slightly more aged in their 30s. In terms of level of education, 56.0% (n = 427) answered that they graduated from university, accounting for more than half of the sample, while for income, 38.3% (n = 292) answered that they earned less than KRW 2 million per month, and 31.6% (n = 241) answered that they earned more than KRW 2 million but less than KRW 3 million per month. As for marital status, unmarried people were the largest percentage with 72.7% (n = 554) (see Table 1).

### 3.2. Descriptive Statistics for Major Variables

Table 2 presents the results of the descriptive statistics for the major variables. First, the average value of Internet overindulgence (the independent variable of this study) was found to be 35.4 (SD = 5.448), while the value for the experience of negative emotion (the moderating variable) was found to be 2.4 (SD = 1.495). In the case of the dysfunctional anger response (the dependent variable), the mean was found to be 2.9 (SD = 0.448), while the functional anger response was found to be 3.1 (SD = 0.468). All skewness (|*s*|< 3) or kurtosis (|*k*|< 10) values did not depart from normality [38].

### 3.3. Analysis of Correlation

Table 3 presents an analysis of the correlation of major variables. All relationships between major variables showed a statistically significant correlation and the direction was positive. The correlation between the dysfunctional anger response and overdependence on the Internet was somewhat high, as the dysfunctional anger response and overdependence on the Internet was r = 0.318, *p* < 0.000, and the functional anger response and overdependence on the Internet was r = 0.184, *p* < 0.000. In the case of negative emotion, as the dysfunctional anger response was r = 0.227, *p* < 0.000 and the functional anger response was r = 0.150, *p* < 0.000, a higher correlation with the dysfunctional anger response was evident as for overdependence on the Internet. Between overdependence on the Internet (the dependent variable) and the experience of negative emotion (the moderating variable), a weak correlation was evident (r = 0.117, *p* < 0.000).

### 3.4. Testing the Hypothesis: Moderating Effect of Negative Emotion

The result of analyzing the moderating effect is shown in Table 4. First, as a result of analyzing the dysfunctional anger response as the dependent variable, the analysis model was found to be statistically significant (F = 38.081, *p* < 0.000, R^2^ = 0.1310). Internet overindulgence (independent variable) and the experience of negative emotion (moderating variable) were found to have a statistically significant positive effect, and the interaction effect of Internet overindulgence × the experience of negative emotion was also found to be statistically significant (coeff = 0.003, t = −1.782, *p* < 0.05, LLCI = −0.0065, ULCI = −0.0003). As the interaction effect was shown to have a positive (+) direction (as with the effectiveness of the dependent variable), it can be seen to strengthen the intensity of the relationship between Internet overindulgence and a dysfunctional anger response. In other words, the empirical result shows that the experience of negative emotion increases the level of dysfunctional anger response caused by Internet overindulgence.

For the functional anger response, only Internet overindulgence showed significance in a positive direction, and the statistical significance of the moderating variable and interaction variable did not appear. In other words, Internet overindulgence had a positive (+) effect, although it also showed statistical significance in that it was found to lead to a functional anger response. As a result, it can be concluded that Internet overindulgence has a significant positive effect on anger response itself, which is the primary concept of dysfunctional and functional anger responses. As a result of analyzing the anger response by dividing it into sub-variables, it can be interpreted that the experience of negative emotion has a significant impact on the amplification of a dysfunctional anger response among the young generation.

Table 5 presents the analysis for the effect on the dysfunctional anger response according to the magnitude of “negative emotion,” i.e., the moderating variable in this study. The change in the slope for Internet overindulgence–dysfunctional anger response according to the magnitude of effectiveness of the moderating effect is shown in Figure 2. When the magnitude of the moderating effect was +1 standard deviation, the slope was relatively steeper. In other words, the higher the level of Internet overindulgence, the higher the level of dysfunctional anger response, and the higher the level of negative emotion, the greater the causal influence of Internet overindulgence–dysfunctional anger response. Thus, this result suggests that alleviating negative emotion among young adults can significantly contribute to lowering the dysfunctional anger response.

## 4. Discussion

Overdependence on the Internet was hypothesized to be a risk factor for a dysfunctional and aggressive anger response and a protective factor for a functional anger response based on the theoretical framework. In accordance with the hypothesis, in this study, Internet overdependence was significantly related to a dysfunctional anger response. Moreover, the moderating effect of negative emotions—that may replace or overlap the emotion of anger provoked by Internet overdependence—was also statistically significant in reducing the probability of a dysfunctional anger response. However, unlike the hypothesis, negative emotions alone significantly predicted a dysfunctional anger response and Internet dependence positively associated with a functional anger response.

Some interpretations may be drawn from these results. First, the frequencies in responding toward anger dysfunctionally and functionally are positively correlated. In this study, the Korean version of the K-BARQ, which consists of six sub-factors, was utilized to measure the degree of functional and dysfunctional anger responses. In a prior study [37] on the validation of the K-BARQ, most sub-factors were also positively correlated. Even though the concepts of functional and dysfunctional anger responses are opposing, people who experience more anger are likely to respond to anger both dysfunctionally and functionally because they have more chances to respond. In this context, the result that Internet dependence positively predicts both dysfunctional and functional anger responses is acceptable.

Second, negative emotions alone were significant predictors of a dysfunctional anger response. In an earlier hypothesis, other negative emotions were assumed to replace anger and deter dysfunctional anger responses. However, without being combined with frustration cues such as Internet dependence, negative emotions are also risk factors for a dysfunctional anger response because these emotions do not act as substitutes for anger.

In this study, some unexpected results were drawn, although the hypothesis was mostly confirmed from the analysis. The unexpected results also provided a new opportunity for understanding the phenomenon. Based on these results, we propose the following implications. First, the validity of the frustration–anger hypothesis was confirmed again in the context of Internet dependence. Second, even in the digital era, Internet dependence may cause frustration triggering aggressive and dysfunctional anger responses. Third, both the K-BARQ and BARQ used to measure the degree of functional and dysfunctional anger responses may have limitations in capturing a negative correlation between functional and dysfunctional anger responses. Fourth, even if it is possible that other negative emotions may replace anger, these emotions can be considered risk factors for dysfunctional anger responses. Fifth, for practitioners who aim to intervene in dealing with anger response problems, engaging with some factors that result in frustration such as Internet dependence may be useful.

### Limitations

The limitations of this study can be summarized as follows: First, the use of traceable longitudinal data was required because the causal relationship was investigated using cross-sectional data. In particular, changes in the phenomenon caused by the COVID-19 pandemic can be said to be a much more important problem. Next, because this study was limited to young adults, it is not possible to demonstrate generational differences. This means the results of this study cannot be generalized to all generations, which has implications for future study development. Finally, this study fails to provide a focus on gender difference. Because aspects of anger response and Internet dependence are identified according to gender, it is necessary to consider this in future studies.

## 5. Conclusions

The aim of this study was to identify the complicated relationship between frustration and anger response during the COVID-19 pandemic. To understand the risk factor and mechanism for dysfunctional anger in the digital era that have been intensified by COVID-19, Internet dependence was selected and the frustration–aggression hypothesis was utilized for the theoretical framework. An analysis was conducted by using an online survey related to addiction during the COVID-19 pandemic. Specifically, this study investigated how overdependence on the Internet affects anger responses (dysfunctional anger, functional anger), and whether negative emotion is shown to have a moderating effect on the relationship between overdependence on the Internet and anger responses. According to the results, the hypothesis based on the theoretical framework was mostly confirmed and some implications were drawn.

When young adults need emotional support, psychological services will provide significant help and will become the major method to prevent extreme consequences. However, even though this study suggests practical implications for fundamentally addressing this issue, bolder policies related to young adults should be developed and implemented by central and local governments. For this, it is necessary to identify problems and perspectives centered on young adults.

## Figures and Tables

**Figure 1 ijerph-20-02435-f001:**
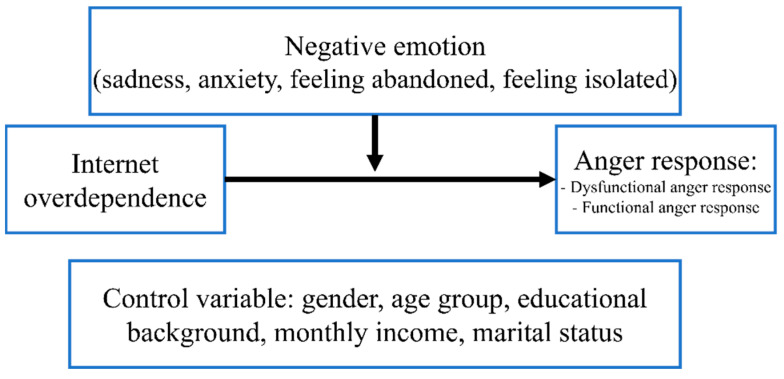
Study model.

**Figure 2 ijerph-20-02435-f002:**
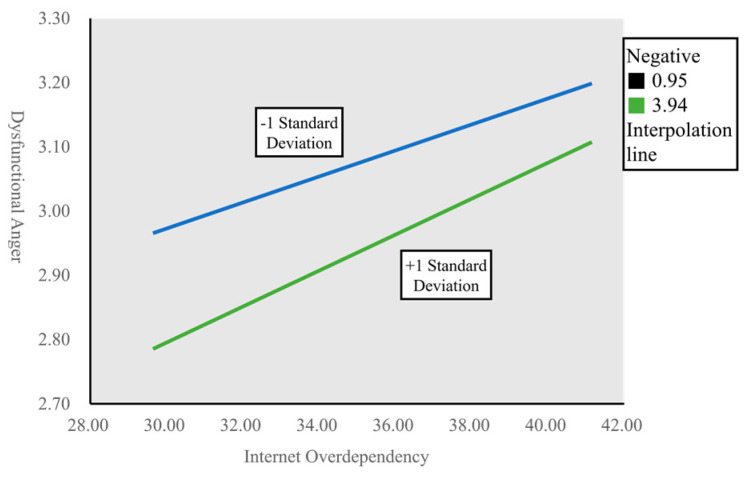
Slope graph of moderating effect.

**Table 1 ijerph-20-02435-t001:** Demographic and sociological characteristics of subjects (N = 762).

		Frequency	Rate
Gender	Male	379	49.7
	Female	383	50.3
Level of Education	<High/school	92	12.1
	Attending college	154	20.2
	Graduate	427	56.0
	>Graduate	73	9.6
	Other	16	2.1
Age	20s	365	47.9
	30s	397	52.1
Monthly Income (KRW)	2 M–3 M	241	31.6
	3 M–4 M	119	15.6
	>400 M	110	14.4
Marital Status	Single	554	72.7
	Married	190	24.9
	Cohabiting	13	1.7
	Divorced	3	0.4
	Separated	2	0.2

**Table 2 ijerph-20-02435-t002:** Descriptive statistics for major variables.

Variable	Minimum Value	Maximum Value	Mean	Standard Deviation	Skewness	Kurtosis
Internet overdependence	15.00	53.00	35.4	5.448	−0.417	0.890
Experience of negative emotion	0.00	5.00	2.4	1.495	−0.015	−0.978
Dysfunctional anger response	1.00	4.49	2.9	0.448	−0.188	1.155
Functional anger response	1.00	4.40	3.1	0.468	−0.692	1.625

**Table 3 ijerph-20-02435-t003:** Analysis of correlation.

	1	2	3
1. Internet overdependence	1	-	-
2. Experience of negative emotion	0.117 *	1	-
3. Dysfunctional anger response	0.318 ***	0.227 ***	1
4. Functional anger response	0.184 ***	0.150 ***	0.338 ***

* *p* < 0.05, *** *p* < 0.001.

**Table 4 ijerph-20-02435-t004:** Testing the hypothesis result of the moderating effect.

Dependent Variable	Variable	Coeff	Se	T	LLCI	ULCI
Dysfunctional anger response	Internet overdependence (X)	0.131	38.081	5.951 ***	1.437	2.141
	Experience of negative emotion (Z)	0.158	0.061	2.563 **	0.037	0.280
	X × Z	0.003	0.001	−1.782 *	−0.007	−0.000
	Invariable	1.789	0.179	9.991 ***	1.437	2.141
R^2^ = 0.131, F = 38.081, *p* = 0.000						
Functional anger response	Internet overdependence (X)	0.019	0.005	3.409 ***	0.008	0.031
	Experience of negative emotion (Z)	0.116	0.067	1.717	−0.017	0.249
	X × Z	−0.002	0.001	−1.196	−0.006	0.002
	Invariable	2.343	0.195	11.966	1.959	2.728
R^2^ = 0.049, F = 12.897, *p* = 0.000						

* *p* < 0.05, ** *p* < 0.01, *** *p* < 0.001.

**Table 5 ijerph-20-02435-t005:** Testing hypothesis result of effectiveness according to the magnitude of moderating variable.

Dependent Variable	Variable	Coeff	se	t	LLCI	ULCI
Dysfunctional anger response	−1 SD	0.019	0.004	7.137	0.011	0.026
	0	0.023	0.003	8.173	0.018	0.029
	+1 SD	0.028	0.004	4.900	0.020	0.036

## Data Availability

Not applicable.

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
