# Peer review of "A Study on the Relationship between Internet Overdependence and Anger Response among Young Adults during COVID-19 Pandemic: Moderating Effect on Negative Emotions"

_ijerph, 2023, doi:10.3390/ijerph20032435_

Round 1

Reviewer 1 Report

Review comments:

This manuscript entitled “A Study on the Relationship between Internet Over-Dependency and Ager among Youth: Moderating Effect on Negative Emotions Concerning COVID-19” aimed to analyses the role of negative emotions caused by COVID-19 on the relationship between overdependence on the Internet and ager that youth people experience in the current situation.

Although this study was read with interest, more work needs to be done in the manuscript to show clearly the potential of the study. In the Introduction Section, it si unclear for me because that insufficient literature on overdependence on the Internet that young people experience in the current situation. On the other hand, why divide it into Introduction section and Background section to write the manuscript? In the Line 182, what is the “00 Institute’s addiction condition survey”?

Author Response

Thank you for your review. We are sending you a response. Please see the attachment.

Reviewer 2 Report

Dear authors:

This is an interesting scientific article, however, it is suggested to consult more relevant references related to studies on the dysfunctional aspects of the Internet and it is recommended to consult the Web of Science and SCOPUS databases.

On the other hand, it is suggested to deepen the analysis of the implications in relation to the young generation in the connection between excessive dependence on the Internet and anger.

It is also recommended to broaden and deepen the conclusions so that they present greater forcefulness in relation to the results obtained.

Finally, it is suggested to further explore the usefulness of the study and its contribution to the scientific community.

Author Response

(The authors gave the same response as above.)

Reviewer 3 Report

1. The introduction provides sufficient background and includes all relevant references.
The introduction speaks right from the specific situation in Korea. It sticks to the theme of "depression" and indeed poses a new problem: the "corona red". There are references to significant previous studies

2. All sources cited are relevant to the research. The 25 citation items and in-house research are sufficient and good.

3. The young man was treated in a new way. He does not serve some "new service", not materially to get "something", but subjectively with rights. I agree that our political governments treat this problem "superficially".
The young man was treated in a new way. Not service, not material, but subjective with rights. I agree that our political governments treat this problem "superficially".

4. The whole methodological side is clear. Sociological norms, all good practices preserved, correlations reliable. Research sample of 762 people good. Of course, I recognize that the authors see that it would be best to: "The limitations of this study can be summarized as follows: First, the use of traceable longitudinal data is required in that the causal relationship was investigated by using crosssectional data. In particular, the tendency of negative emotions caused by COVID-19 can be said to be a much more important problem" (317-320).

5. All the results and termination are well elaborated.

I fully support the printing of the scientific article in its entirety.

Author Response

(The authors gave the same response as above.)

Reviewer 4 Report

Title

A Study on the Relationship between Internet Over-Dependency and Anger among Youth: Moderating Effect on Negative Emotions Concerning COVID-19

In the title, youth is vague.

Abstract

This part is more than an abstract, it is an expression of the problem.  The authors haven’t reported the objective, method, result, and conclusion.

Method

-             Each of the measures should be carefully presented, with the validity and reliability indexes for each of them for the original version and the Korean version.

-             The data used in this study is secondary, it is necessary to provide information from the primary data, which includes how many subjects and what percentage of the subjects were included in this study, there isn’t any information about the population, sampling method, sample selection, and inclusion and exclusion criteria.

-             The procedure of the study hasn’t been reported.

Findings

-             In the results section, figure 1, authors have said that they have controlled the effect of all demographic variables (gender, age group, education and ….), while a variety of different classes is observed for such variables. Of course, there may have been statistical control, which should be mentioned exactly, how much control was applied. Report results with and without statistic controls for these variables.

-             In Table 2- The kurtosis index for both components of anger is very high and this may be significantly different from the normal distribution, it is necessary to report the Kolmogorov–Smirnov statistic, if there is a difference between the observed distribution and the normal distribution, the statistical methods will also be affected.

-             The presented coefficients, to predict anger from mobile phone addiction, it is necessary to include the contribution of moderators and controls.

-             Provide direct and indirect coefficients by considering moderating effect of COVID-19

-             How to confirm the presence or absence of the Moderating Effect in the model?

Author Response

(The authors gave the same response as above.)

Round 2

Reviewer 1 Report

In the revised manuscript, the authors substantially improve the presentation of the contributions and the experimental results of their proposed issues according to my comments. In my opinion, the revised version of the manuscript is now acceptable for publication. 

Author Response

(The authors gave the same response as above.)

Reviewer 4 Report

The participants are from different ages, gender, and educational groups; therefore, these variables cannot be controlling variables. It needs to analyze and report the results according to the confounding contribution of these variables.
Please report the direct and indirect coefficients according to the role of mediating variables.

Author Response

(The authors gave the same response as above.)
